# Characteristics of aldosterone-producing adenomas in patients without plasma renin activity suppression

Haremaru Kubo[1], Yuya Tsurutani[1]*, Kosuke Inoue[1,2], Kazuki Watanabe[1], Yuto Yamazaki[3], Takashi Sunouchi[1], Yoshitomo Hoshino[1], Rei Hirose[1], Sho Katsuragawa[1], Hiromitsu Tannai[4], Yukiko Shibahara[5,6], Yukio Kakuta[5], Seishi Matsui[4], Jun Saito[1], Masao Omura[1], Hironobu Sasano[3], Tetsuo Nishikawa[1]

1 Endocrinology and Diabetes Center, Yokohama Rosai Hospital, Yokohama, Kanagawa, Japan,
2 Department of Social Epidemiology, Graduate School of Medicine, Kyoto University, Kyoto, Japan,
3 Department of Pathology, Tohoku University Graduate School of Medicine, Sendai, Miyagi, Japan,
4 Department of Radiology, Yokohama Rosai Hospital, Yokohama, Kanagawa, Japan, 5 Department of Pathology, Yokohama Rosai Hospital, Yokohama, Kanagawa, Japan, 6 Laboratory Medicine and Pathobiology, University of Toronto, Toronto, Ontario, Canada

* yuya97tsuru1055@gmail.com

**Data Availability Statement:** All relevant data are within the paper and its Supporting Information Files.

## Abstract

Primary aldosteronism (PA) usually accompanies suppressed plasma renin activity (PRA) through a negative feedback mechanism. While some cases of PA with unsuppressed PRA were reported, there have been no studies about the characteristics of PA with unsuppressed PRA; thus, these characteristics were examined herein. Nine patients with unsuppressed PRA and 86 patients with suppressed PRA were examined. All patients underwent segmental adrenal venous sampling (sAVS) and adrenalectomy, and were pathologically confirmed to have cytochrome P450 11B2 (CYP11B2)-positive aldosterone-producing adenoma according to international histopathology consensus criteria. Unsuppressed and suppressed PRA were defined as PRA levels of > 1.0 and $\leq$ 1.0 ng/mL/hr, respectively, in multiple blood samples obtained in the resting position. The unsuppressed PRA group had higher morning cortisol levels (12.6 [8.5, 13.5] vs. 8.5 [7.1, 11.0] μg/dL, $P = 0.03$) and higher cortisol levels after a 1 mg dexamethasone suppression test (DST) (2.2 [1.6, 2.5] vs. 1.3 [1.0, 1.9] μg/dL, $P = 0.004$) than the suppressed PRA group. The unsuppressed PRA group also showed higher aldosterone levels on the non-surgical side during sAVS ($P = 0.02$ before adrenocorticotropic hormone (ACTH) stimulation, $P = 0.002$ after ACTH stimulation), a higher intensity of CYP17 expression in the resected adrenal gland ($P = 0.02$), and a lower clinical complete success rate 1 year after surgery ($P = 0.04$) compared with those in the suppressed PRA group. These findings suggest that PA should not be ruled out by unsuppressed PRA among patients with hypertension, particularly when their cortisol levels remain unsuppressed in the 1 mg DST. Meanwhile, it should be acknowledged that patients with unsuppressed PRA have higher aldosterone levels on the non-surgical side, and a lower likelihood of postoperative complete clinical success is to be expected.

**Funding:** The authors received no specific funding for this work.

**Competing interests:** The authors have declared that no competing interests exist.

## Introduction

Primary aldosteronism (PA), characterized by excessive production of aldosterone from the adrenal glands, is the most common cause of secondary hypertension and is thought to be underdiagnosed [1]. In addition to a high plasma aldosterone concentration (PAC), patients with PA usually have suppressed plasma renin activity (PRA) owing to an aldosterone-driven negative feedback mechanism. Hence, measuring the PAC (ng/dL)/PRA (ng/mL/hr) ratio (aldosterone-to-renin ratio [ARR]) is recommended as a screening test for PA; the Japanese Endocrine Society (JES) guidelines set the cut-off for PA at an ARR of > 20 [2]. In addition to increased ARR, some investigators include PRA suppression (i.e., < 1.0 ng/mL/hr) as a screening criterion [3]; however, hypertensive patients with unsuppressed PRA might not be considered to have PA based on this criterion. Given that PAC is reported to cause organ failure (especially left ventricular mass) in patients with PA independently of PRA [4], PAC and PRA may have different mechanisms of influencing metabolic regulation. This suggests that higher PAC levels may be harmful to human health even if the PRA is not suppressed.

Although there are a few case reports of patients who have PA with unsuppressed PRA [5–7] due to renal artery stenosis [6] or the development of hypertensive nephrosclerosis [7], information regarding the characteristics of such patients remains unclear. In such cases, deviating from the guidelines, screening tests and confirmatory tests using ARR are thought to be relatively negative because of elevated PRA. Thus, clinical phenotypes (for example, uncontrollable hypertension and the existence of hypokalemia and adrenal tumor) may be keys for further study. In this study, we investigated the clinical characteristics of patients with PA who had unsuppressed PRA and compared them with those with suppressed PRA.

## Materials and methods

### Patient population and data collection

We retrospectively reviewed our clinical records and extracted the patients with aldosterone-producing adenoma (APA) in whom PRA was unsuppressed (n = 9) and suppressed (n = 86). All patients met the following inclusion criteria: 1) patients who underwent segmental adrenal venous sampling (sAVS) [8–10] and had confirmed excess aldosterone on sAVS between 2007 and 2020 at our hospital; 2) patients who underwent adrenalectomy and immunohistochemical evaluation of multiple steroidogenic enzymes (11β-hydroxylase cytochrome P450 (CYP11B1), aldosterone synthase cytochrome P450 (CYP11B2), 17-alpha-hydroxylase (CYP17), 3-β hydroxysteroid dehydrogenase-isomerase 1 (HSD3B1), 3-β hydroxysteroid dehydrogenase-isomerase 2 (HSD3B2), and dehydroepiandrosterone sulfotransferase (DHEA-ST)) in the resected adrenal gland; and 3) patients who were diagnosed with APA in which CYP11B2 expression was confirmed. All patients provided written informed consent for performing the immunohistochemistry. Patients who did not undergo surgery were excluded because they were not evaluated histologically.

Unsuppressed and suppressed PRA were defined as PRA levels of > 1.0 and ≤ 1.0 ng/mL/hr in any two or more blood samples obtained in the resting position, respectively. Patients with suppressed and unsuppressed PRA (for example, PRA levels of > 1.0 ng/mL/hr in two confirmatory tests and ≤ 1.0 ng/mL/hr in one confirmatory test) were excluded from the analysis.

This retrospective study protocol was approved by the Research Ethics Committee of Yokohama Rosai Hospital (approval no. 30–46).

## PAC, PRA, serum cortisol concentration, and confirmatory test determination

We optimized the prescription of antihypertensive drugs to patients several weeks before blood sampling according to the JES guidelines [2]. Particularly, patients treated with mineralocorticoid receptor antagonists and diuretics stopped taking these drugs for longer than 4 weeks before admission. Patients who showed hypokalemia were treated with potassium supplements during this period. Patients were served a normal diet without sodium restriction (routinely 8 g/day), and data were obtained in consideration of a plausible condition; for example, without exposure to caffeine and extensive stress [11]. All blood pressure measurements were conducted by an attending physician to ensure adequate and proper pre- and post-operative assessment [12] according to The Japanese Society of Hypertension guidelines [13]. Morning blood samples were collected after the patients had rested for 30 minutes. PACs, serum cortisol concentrations, and PRAs were measured using radioimmunoassays as previously described [14, 15] and standardly used [9, 10, 16].

Because some studies have shown that confirmatory tests such as the sodium loading test or the furosemide loading test may be dangerous, especially in those with a pre-existing condition that increases risk, such as severe hypokalemia, hypertension, and heart failure [17], patients with such comorbidities were exempted from some confirmatory tests. The screening tests for PA were conducted based on the JES guidelines [2]. Patients with typical phenotypes such as treatment-resistant hypertension or hypokalemia, or those with positive results in the screening tests (including an ARR $\geq$ 20 or outpatients with one or more positive results in confirmatory tests conducted in other hospitals) were admitted to our hospital for further examination. In the suppressed PRA group, diagnosis of PA was determined in accordance with the JES guidelines as positive results in one or more of the confirmatory tests [2]. In the unsuppressed PRA group, the patients' phenotypes or confirmatory test results were carefully considered, and further examination was conducted once adequate informed consent was provided. Excessive aldosterone levels were confirmed via sAVS, and the pathological characteristics are described in later sections. Undetectable PRA levels ($< 0.1$ ng/mL/hr) were considered as 0.1 ng/mL/hr in the statistical analyses.

## AVS and imaging studies

We performed thin-section computed tomography (CT) of the adrenal glands after administering an intravenous injection of contrast medium as well as sAVS to determine the laterality of each patient's hyperaldosteronism. We considered the adrenal vein to be affected by aldosterone hypersecretion when the effluent aldosterone concentrations before and 15–90 minutes after ACTH stimulation were $> 250$ ng/dL and $> 1,400$ ng/dL, respectively, as described previously [8–10, 18]. The lateralization index (LI) and contralateral ratio (CR) were calculated as (PAC/cortisol in the central vein of the resected adrenal gland)/(PAC/cortisol in the central vein of the unresected adrenal gland) and (PAC/cortisol in the central vein of the unresected adrenal gland)/(peripheral PAC/cortisol) after ACTH stimulation, respectively [19].

## Pathological examination procedure

All patients were diagnosed with benign adrenocortical adenoma based on the histological criteria of Weiss [20]. After precisely reviewing the morphological findings on hematoxylin and eosin (HE)-stained tissues for all cases, we conducted immunohistochemical analysis of the following steroidogenic enzymes: CYP11B2, CYP11B1, CYP17, HSD3B1, HSD3B2, and DHEA-ST [21–24]. Aldosterone production in the tumor was assessed based on the

HISTALDO consensus [25]. Immunoreactivity was assessed depending on both the immunointensity and distribution of the specific immunoreactivity using McCarty's H-scoring system, in which the percentage of stained cells is multiplied by a number reflecting the immunopositive staining intensity, as described previously [26].

### Determination and evaluation of clinical and outcomes

Clinical outcomes 1 year after surgery were assessed according to the Primary Aldosteronism Surgical Outcomes (PASO) criteria [27]. Biochemical outcomes were excluded from this analysis because elevated PRA can make it difficult to evaluate ARR normalization and to perform confirmatory tests.

### Statistical analysis

Continuous variables are shown as the median (interquartile range) for non-normal distributions. The PRA and PAC of each patient in the unsuppressed group (2–3 datapoints each) are shown as mean values. The differences in clinical parameters between the unsuppressed and suppressed PRA groups were analyzed using the Mann-Whitney U test for continuous variables or Fisher's exact test for categorical variables unless otherwise noted. The analyses were performed using JMP software, version 12.0.1 (SAS Institute Inc., Cary, NC, USA).

## Results

### Classification of the unsuppressed and suppressed PRA groups and assessment of clinical parameters

Among the patients who visited our hospital between 2007 and 2020, nine and 86 patients fulfilled the inclusion criteria for unsuppressed and suppressed PRA in this study, respectively. The baseline characteristics and endocrine parameters of each patient with unsuppressed PRA are shown in S1 and S2 Tables, respectively. Four patients did not fulfill the screening criteria for a PA diagnosis according to the JES guidelines [2] (ARR > 20), as shown in S2 Table, probably due to elevated PRA levels. However, considering they exhibited typical and clinically important characteristic signs of PA (uncontrollable hypertension, hypokalemia, or adrenal tumor) and their strong wishes, we carefully proceeded with further examination after being provided adequate informed consent. All patients except patient 3 had excess aldosterone, as observed from the results of the saline loading test [28]. The results of the furosemide loading test and the captopril loading test were frequently negative, probably due to unsuppressed PRA. For these reasons, four patients (patients 1, 2, 3, and 5) did not meet the criteria outlined in the JES guidelines for a PA diagnosis. No patient had renal artery stenosis, and all patients except patient 2 had an adrenal tumor, as noted on a multi-slice helical contrast-enhanced CT scan. Five patients had cortisol levels ≥ 1.8 μg/dL, which is the cut-off for subclinical Cushing's syndrome after the 1 mg dexamethasone suppression test (DST) [29].

A comparison of the clinical characteristics of the unsuppressed and suppressed PRA groups is shown in Table 1; there were no differences in sex, age, body mass index (BMI), or serum potassium levels between the two groups. The frequency of diabetes mellitus (DM), including borderline diabetes, was significantly higher in the unsuppressed PRA group than in the suppressed group (55.6% vs. 16.3%, $P = 0.01$). The estimated glomerular filtration rate (eGFR), urine albumin level, and brachial-ankle pulse wave velocity were not significantly different between the two groups.

Endocrine parameters of the two groups are shown in Table 2. PAC and urine aldosterone values were not significantly different between the two groups. The morning cortisol level and

**Table 1. Comparison of patients' clinical characteristics between the unsuppressed and suppressed PRA groups[a].**

| | Unsuppressed PRA Group | Suppressed PRA Group | P-value |
|---|---|---|---|
| | (n = 9) | (n = 86) | |
| Sex (male/female) | 4/5 | 35/51 | 0.82 |
| Age (years) | 51.0 [46.0, 56.5] | 50.5 [42.0, 60.0] | 0.81 |
| BMI (kg/m$^2$) | 24.9 [19.2, 28.4] | 23.1 [21.1, 25.5] | 0.99 |
| Systolic blood pressure (mmHg) | 133.0 [124.5, 152.5] | 148.0 [134.0, 167.3] | 0.07 |
| Diastolic blood pressure (mmHg) | 89.0 [78.0, 99.5] | 90.0 [80.0, 102.5] | 0.56 |
| Duration of hypertension (years) | 10 [8, 19] | 8 [2, 16] | 0.15 |
| History of cardiovascular disease | 2 (22.2%) | 12 (14.0%) | 0.51 |
| Diabetes mellitus[b] | 5 (55.6%) | 14 (16.3%) | 0.01 |
| Dyslipidemia | 3 (33.3%) | 20 (23.3%) | 0.50 |
| Number of antihypertensive drugs | 2 [1, 2] | 2 [1, 2] | 0.64 |
| Serum potassium (mEq/L) | 3.2 [2.9, 3.6] | 3.2 [2.9, 3.5] | 0.90 |
| Oral potassium supplements, n (%) | 6 (66.7%) | 53 (61.6%) | 0.76 |
| eGFR (mL/min/1.73 m$^2$) | 80.3 [51.5, 91.5] | 79.5 [67.9, 99.5] | 0.15 |
| Urinary sodium (mEq/day) | 137.6 [130.1, 172.0] | 151.2 [116.8, 188.6] | 0.69 |
| Urine albumin (mg/g･Cr) | 16.7 [12.6, 31.1] | 13.6 [7.5, 37.5] | 0.18 |
| baPWV (m/sec) | 1498.5 [1264.9, 1641.6] | 1514.5 [1350.6, 1696.9] | 0.20 |
| Tumor size by CT scan (mm) | 14 [9.5, 23] | 13.8 [10, 16] | 0.80 |
| Laterality, right/left | 5/4 | 43/43 | 0.75 |

PRA, plasma renin activity; BMI, body mass index; eGFR, estimated glomerular filtration rate; baPWV, brachial-ankle pulse wave velocity; CT, computed tomography.

[a]Continuous variables are shown as median [interquartile range] for non-normal distributions.

[b]Diabetes mellitus included borderline diabetes mellitus.

the level after the 1 mg DST were significantly higher in the unsuppressed PRA group than in the suppressed PRA group (12.6 [8.5, 13.5] vs. 8.5 [7.1, 11.0] μg/dL, *P* = 0.03, and 2.2 [1.6, 2.5] vs. 1.3 [1.0, 1.9] μg/dL, *P* = 0.004, respectively). The number of patients exhibiting a higher cortisol level after the administration of 1 mg DST ($\geq$ 1.8 μg/dL) tended to be higher in the unsuppressed group than in the suppressed group (55.6% vs. 26.7%, respectively; *P* = 0.07).

The results and interpretation of sAVS data in the unsuppressed group are shown in S3 Table. A comparison of the results of sAVS of the two groups is shown in Table 3. LI was lower and CR was significantly higher in the unsuppressed PRA group than in the suppressed PRA group (3.4 [1.48, 8.45] vs. 10.1 [2.5, 26.2], *P* = 0.03, and 0.60 [0.41, 1.91] vs. 0.20 [0.12, 0.55], *P* = 0.01, respectively). PAC and cortisol levels in the central vein of the surgical side before and after ACTH stimulation were not significantly different between the two groups. Maximum PAC and cortisol levels in the tributary vein were also not different between the two groups. Meanwhile, PAC in the central vein of the non-surgical side before and after ACTH stimulation was higher in the unsuppressed PRA group than in the suppressed group (237.5 [79.1, 653.3] vs. 66.8 [40.9, 135.8] ng/dL, *P* = 0.02, and 1420.0 [739.5, 2510.0] ng/dL vs. 423.0 [304.0, 785.0] ng/dL, *P* = 0.002, respectively).

## Immunohistochemistry and immunohistochemical characterization

Several patients in the unsuppressed group did not fulfill the screening or diagnostic criteria for PA, but CYP11B2 expressions were confirmed in all patients (S1 Fig). Similarly, CYP11B2 expression was also confirmed in all patients in the suppressed group. Representative histological images of APA of the two groups are shown in Fig 1, and the qualitative evaluation is

**Table 2. Comparison of endocrine parameters in the unsuppressed and suppressed PRA groups[a].**

| | Unsuppressed PRA Group (n = 9) | Suppressed PRA Group (n = 86) | P-value |
|---|---|---|---|
| PRA (ng/mL/hr) | 1.8 [1.5, 2.5] | 0.2 [0.1, 0.4] | <0.001 |
| PAC at 8:00 (ng/dL) | 34.3 [18.5, 102.5] | 27.1 [20.0, 41.0] | 0.40 |
| Cortisol at 8:00 (μg/dL) | 12.6 [8.5, 13.5] | 8.5 [7.1, 11.0] | 0.03 |
| Cortisol at 23:00 (μg/dL) | 3.4 [2.1, 4.5] | 2.8 [2.0, 3.6] | 0.29 |
| ACTH at 8:00 (pg/mL) | 14.3 [12.2, 26.8] | 15.9 [11.4, 23.3] | 0.99 |
| ACTH at 23:00 (pg/mL) | 8.7 [4.8, 9.3] | 8.3 [4.6, 13.9] | 0.43 |
| Urinary aldosterone (μg/day) | 18.5 [14.3, 28.9] | 19.9 [11.6, 34.5] | 0.99 |
| Urinary cortisol (μg/day) | 48.3 [34.8, 69.0] | 42.4 [33.0, 58.1] | 0.55 |
| PAC (240 min after saline loading) | 36.2 [19.1. 70.3] | 21.5 [12.5, 39.1] | 0.10 |
| PRA (120 min after furosemide loading) | 3.4 [3.1, 5.6] | 0.3 [0.2, 0.7] | <0.001 |
| ARR (90 min after captopril loading) [b] | 15.3 [5.8, 48.3] | 156.0 [74.0, 330.0] | <0.0001 |
| Max PAC/cortisol ratio after ACTH stimulation | 2.7 [2.1, 6.9] | 2.8 [2.0, 4.3] | 0.60 |
| Cortisol level after 1 mg DST (μg/dL) | 2.2 [1.6, 2.5] | 1.3 [1.0, 1.9] | 0.004 |
| Number of patients with higher cortisol level (≥ 1.8 μg/dL) after 1 mg DST | 5 (55.6%) | 23 (26.7%) | 0.07 |

PRA, plasma renin activity; PAC, plasma aldosterone concentration; ACTH, adrenocorticotropic hormone; ARR, aldosterone/renin ratio; DST, dexamethasone suppression test.

Conversion to SI units: PAC, ng/dL × 27.7 for pmol/L; Cortisol, μg/dL × 27.6 for nmol/L; ACTH, pg/mL × 0.220 for pmol; Urinary aldosterone, μg/day × 2.77 for nmol/day; Urinary cortisol, μg/day × 2.76 for nmol/day.

[a]Continuous variables are shown as median [interquartile range] values for non-normal distributions.

[b]ARR is calculated as PAC (ng/dL) divided by PRA (ng/mL/hr).

**Table 3. Results of segmental adrenal venous sampling in the unsuppressed and suppressed PRA groups[a].**

| | Unsuppressed PRA Group (n = 9) | Suppressed PRA Group (n = 86) | P-value |
|---|---|---|---|
| LI | 3.4 [1.5, 8.5] | 10.1 [2.5, 26.2] | 0.03 |
| CR | 0.6 [0.4, 1.9] | 0.2 [0.1, 0.6] | 0.01 |
| Surgical side | | | |
| Cortisol in the central vein before ACTH | 160.0 [33.9, 678.8] | 113.0 [41.6, 352.0] | 0.59 |
| Cortisol in the central vein after ACTH | 785.0 [585.0, 1100.0] | 750.0 [578.0, 943.5] | 0.54 |
| PAC in the central vein before ACTH | 2020.0 [328.7, 3110.0] | 1590.0 [369.0, 4190.0] | 0.81 |
| PAC in the central vein after ACTH | 3020.0 [1420.0, 7060.0] | 5440.0 [2255.0, 11050.0] | 0.36 |
| maxCortisol in the tributary vein after ACTH | 1020.0 [611.0, 1243.0] | 909.5 [769.0, 1155.0] | 0.77 |
| maxPAC in the tributary vein after ACTH | 14100.0 [3235.0, 24476.8] | 9910.0 [5222.5, 23500.0] | 0.81 |
| Non-surgical side | | | |
| Cortisol in the central vein before ACTH | 93.8 [56.9, 311.7] | 112.5 [31.2, 295.5] | 0.80 |
| Cortisol in the central vein after ACTH | 547.5 [476.2, 832.8] | 683.0 [521.0, 879.0] | 0.53 |
| PAC in the central vein before ACTH | 237.5 [79.1, 653.3] | 66.8 [40.9, 135.8] | 0.02 |
| PAC in the central vein after ACTH | 1420.0 [739.5, 2510.0] | 423.0 [304.0, 785.0] | 0.002 |

PRA, plasma renin activity LI, lateralization index; CR, contralateral ratios; ACTH, adrenocorticotropic hormone; PAC, plasma aldosterone concentration; PRA, plasma renin activity.

Conversion to SI units: PAC, ng/dL × 27.7 for pmol/L; Cortisol, μg/dL × 27.6 for nmol/L.

[a]Continuous variables are shown as median [interquartile range] for non-normal distributions.

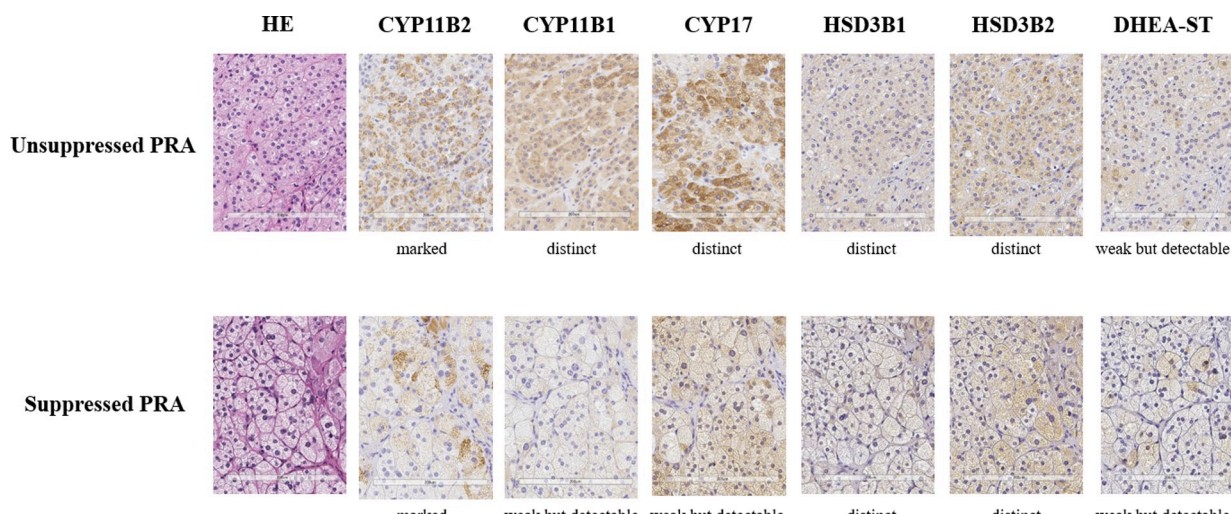

**Fig 1. Representative histological images of aldosterone-producing adenomas with unsuppressed and suppressed PRA.** HE, hematoxylin and eosin; CYP11B1, 11β-hydroxylase cytochrome P450; CYP11B2, aldosterone synthase cytochrome P450; CYP17, 17alpha-hydroxylase; HSD3B1, 3-β hydroxysteroid dehydrogenase-isomerase 1; HSD3B2, 3-β hydroxysteroid dehydrogenase-isomerase 2; DHEA-ST, dehydroepiandrosterone sulfotransferase. Immunoreactivity of CYP17 was more abundant in the unsuppressed PRA group. CYP11B2 immunoreactivity was strongly detected, and CYP11B1 immunoreactivity was distinctly detected in both groups. Scale bar: 200 μm.

summarized in Table 4. Cortical atrophy of the attached normal cortex was not detected in the unsuppressed PRA group. The intensity of CYP17 labeling was higher in the unsuppressed

**Table 4. Histopathological characteristics in the unsuppressed and suppressed PRA groups.**

| | Unsuppressed PRA Group[b] | Suppressed PRA Group | P-value[a] |
|---|---|---|---|
| | (n = 9) | (n = 86) | |
| Dominant cell type | | | |
| Clear | 8 | 69 | 1.00 |
| Compact | 0 | 9 | 0.59 |
| Mixed | 1 | 4 | 0.40 |
| CYP11B2 | | | |
| Absent or weak/distinct or very strong | 0/9 | 3/83 | 0.57 |
| CYP11B1 | | | |
| Absent or weak/distinct or very strong | 3/6 | 47/38 | 0.21 |
| CYP17 | | | |
| Absent or weak/distinct or very strong | 1/8 | 45/40 | 0.02 |
| HSD3B1 | | | |
| Absent or weak/distinct or very strong | 4/5 | 38/43 | 0.89 |
| HSD3B2 | | | |
| Absent or weak/distinct or very strong | 4/5 | 24/57 | 0.36 |
| DHEA-ST | | | |
| Absent or weak/distinct or very strong | 6/3 | 57/24 | 0.81 |

PRA, plasma renin activity; CYP11B1, 11β-hydroxylase cytochrome P450; CYP11B2, aldosterone synthase cytochrome P450; CYP17, 17alpha-hydroxylase; HSD3B1, 3-β hydroxysteroid dehydrogenase-isomerase 1; HSD3B2, 3-β hydroxysteroid dehydrogenase-isomerase 2; DHEA-ST, dehydroepiandrosterone sulfotransferase.

[a]Parameters were analyzed using Fisher's exact test.

[b]Some specimens to be analyzed in each category in the suppressed PRA group were missing. Missing samples: dominant cell type: 4; CYP11B1: 1; HSD3B1d: 5; HSD3B2: 5; and DHEA-ST: 5.

PRA group than in the suppressed PRA group (absent or weak/distinct or very strong; 1/8 vs. 45/40, P = 0.02). The intensities of the labeling of other steroidogenic enzymes were not significantly different between the two groups.

## Assessment of postoperative outcomes

Clinical outcomes 1 year after surgery were assessed according to the PASO criteria [27] are shown in Table 5. This standardized outcome criteria means remission (complete success), improvement (partial success), and persistence (absent success) of PA after adrenalectomy from the viewpoint of both clinical (mainly about blood pressure and the existence of medications) and biochemical outcomes (mainly about excess of aldosterone and renin suppression), respectively. Clinically, Partial or complete clinical success was achieved in 8 patients in the unsuppressed PRA group. Complete success rate was lower in the unsuppressed PRA group than in the suppressed PRA group (11.1% vs. 48.1%, P = 0.04).

In the unsuppressed group, biochemical outcomes requiring ARR normalization could not be evaluated because of the elevated PRA before the surgery. However, correction of hypokalemia, one of the criteria of the biochemical outcome assessment, was confirmed in all patients who required potassium supplementation before surgery.

## Discussion

Herein, we presented nine patients with APA who had no PRA suppression and summarized their characteristics. We also found a higher frequency of patients in the unsuppressed PRA group who exhibited unsuppressed cortisol levels after the 1 mg DST. Suppressed PRA and higher PAC (in the context of a suppressed PRA phenotype) are associated with the incidence of hypertension, even among individuals who have no hypertension (i.e., subclinical primary aldosteronism) [30]. However, little is known about the unsuppressed PRA phenotype with hypertension caused by excess aldosterone. Moreover, previous studies rarely reported the potential contribution of the ACTH-cortisol pathway in the genesis of PA [31, 32]. In this context, our findings provide new evidence about the clinical characteristics of patients with unsuppressed PRA, indicating the importance of careful consideration of patients with PA without PRA suppression.

Patients in the unsuppressed PRA group in our study had higher morning cortisol levels and higher levels after the 1 mg DST than those in the suppressed PRA group. In addition, we found that the intensity of CYP17 expression in the resected adenomas was higher in the unsuppressed PRA group. It is reported that 10–20% of patients with APA also have subclinical Cushing's syndrome [33–35], and a relatively high percentage of APAs comprise cells that

**Table 5. Clinical outcomes 1 year after the surgery in the unsuppressed and suppressed PRA groups[a].**

|  | Unsuppressed PRA Group | Suppressed PRA Group | P-value[bc] |
|---|---|---|---|
| Complete success rate | 1/9 (11.1%) | 37/77 (48.1%) | 0.04 |
| Complete success | 1 | 37 | |
| Partial success | 7 | 36 | |
| Absent success | 1 | 4 | |
| Unevaluable | 0 | 9 | |

PRA, plasma renin activity.

[a]Clinical outcomes were evaluated according to the Primary Aldosteronism Surgical Outcomes criteria [27].

[b]Nine patients in the suppressed group were missing and unevaluable for clinical parameters.

[c]Parameters were analyzed using Fisher's exact test.

are double-positive for CYP11B1 and CYP17, which are key enzymes in cortisol biosynthesis [21, 23, 36]. Cortisol production from the adenoma may elevate PRA levels by increasing the renin substrate angiotensinogen [37] while suppressing aldosterone secretion through the ACTH pathway [32]. In fact, elevated PRA and renin substrate levels in patients with cortisol-producing adenoma (CPA) have been previously reported [38, 39]. Additionally, our findings are consistent with a study that showed increased PRA levels and a normal ARR in patients with PA who had CPAs that excluded the patient from further screening tests [40]. However, many of the patients in the unsuppressed PRA group with cortisol levels $\geq$ 1.8 μg/dL after the 1 mg DST did not have suppressed ACTH secretion, nor did they exhibit a disruption of the diurnal rhythm of cortisol levels; thus, it is unclear how cortisol secretion affected the condition. In addition, the difference between the unsuppressed and suppressed PRA groups was not significant for other factors, including the levels of ACTH, urinary cortisol, and cortisol measured via sAVS, as well as CYP11B1 expression. Moreover, not all patients with subclinical hypercortisolism exhibited elevated PRA levels [41]. Further investigation is needed to identify the possible relationship between unsuppressed PRA and higher cortisol secretion among patients with PA.

With regard to glucose metabolism, the higher prevalence of diabetes in the unsuppressed PRA group should be considered as a factor that can contribute to elevated PRA levels. This possibility is supported by the fact that diabetic complications could activate the renin-angiotensin system by accelerating atherosclerosis and chronic kidney disease [42, 43]. Additionally, a higher frequency of DM in the unsuppressed PRA group also might be derived from subclinical hypercortisolism [44].

Furthermore, it has been reported that younger age [45], decreased eGFR [46] and sodium intake [47], and renal artery stenosis [48] are associated with elevated PRA levels. However, as shown in Table 1, there were no significant differences between the two groups in terms of age, eGFR, and 24-hour urinary sodium excretion, which is a biomarker of sodium intake. Additionally, there were no patients who exhibited renal artery stenosis in the unsuppressed PRA group, as detected on contrast-enhanced multi-slice helical CT. Focusing on each patient, four of nine patients (patients 1, 4, 5, and 7) in the unsuppressed PRA group had chronic kidney disease (defined as an eGFR less than 60 mL/min/1.73 m$^2$) or a urine albumin-to-creatinine ratio (UACR) > 30 mg/g creatinine [49]. In addition, the duration of hypertension was relatively longer in patients 1, 2, 5 and 7, which might have led to elevation of PRA via the activation of the renin-angiotensin system [50]. These factors might be associated with an elevation of PRA. Additional investigations with a larger sample size and sufficient statistical power are required for confirmation.

Partial or complete clinical success was achieved in eight out of nine patients in the unsuppressed PRA group; however, the complete success rate was lower in the unsuppressed PRA group than in the suppressed PRA group. As shown in S3 Table, excess aldosterone in the central or tributary vein in the unresected side was observed in four patients (patients 1, 4, 5 and 9). Additionally, in these cases, some lesions associated with excess aldosterone (such as small-sized lesions of an aldosterone-producing nodule, multiple aldosterone-producing nodules (MAPN) or micronodules, or aldosterone-producing diffuse hyperplasia) [10, 25, 36, 51–53] might exist on the unresected side. Histological analysis revealed that only patient 1 had MAPN in the resected adrenal gland. Alternatively, aldosterone values on the non-surgical side might be increased by unsuppressed PRA. Predictors for clinical outcomes in a previous study (such as sex, BMI, duration of hypertension, number of antihypertensive drugs, and tumor size [54]) did not differ between the two groups. Otherwise, from the viewpoint of hypokalemia correction, all the patients with pre-existing hypokalemia (patients 1, 2, 6, 7, 8 and 9) were spared from requiring oral potassium supplements suggesting partial biochemical

improvement, though we could not evaluate biochemical success because of decreased pre-operative ARR due to elevated PRA in the unsuppressed PRA group. When surgery is planned on a patient with unsuppressed PRA, the possibility of an incomplete cure of hypertension should be well-explained.

The diagnosis of PA in patients with unsuppressed PRA may be difficult. In daily practice, the possibility of PA may be ruled out if a patient with hypertension has unsuppressed PRA. Actually, four out of nine patients in our study did not meet the criteria outlined in the JES guideline for a PA diagnosis because their screening tests were negative due to elevated PRA (ARR < 20) [2]. We performed detailed examinations of our patients with adequate informed consent because they had adrenal tumors, hypokalemia, or a high insistence on being treated alongside other complications, including diabetes. In terms of the diagnosis, it also should be noted that five out of five patients had negative furosemide plus upright test results, and four out of nine patients had negative captopril loading test results; these tests tend to be negative since they include PRA levels in their criteria. On the other hand, the saline-loading test may be useful because it does not incorporate PRA levels in its criteria; in fact, eight out of eight patients had positive saline-loading test results. However, this test should not be performed in patients for whom it is considered unsafe; for example, in uncontrolled hypertension. Indeed, some patients were exempted from these confirmatory tests in the unsuppressed PRA group. For such patients, especially those with a CT-detectable adrenal tumor, AVS may be one of the useful tools for the detection of excess aldosterone, although the application should be considered carefully. Moreover, the patients with unsuppressed PRA in this study might represent a heterogenous group; for example, they exhibited a broad range of PAC measurements in the confirmatory tests. This could be derived from each patient's comorbidities. For example, patients 1 and 5–9 had higher cortisol levels after the 1 mg DST; patients 1, 4, and 5 had chronic kidney disease (eGFR < 60 mL/min/1.73 m$^2$); and patients 1, 3–5, and 7 had diabetes. Furthermore, patient 2 had a history of stroke and a long duration of hypertension (21 years), suggesting advanced atherosclerosis. These multiple factors could have contributed to the elevation of PRA and PAC, and further investigations with more patients might reveal the associations of each factor.

There were several limitations in this study. First, its statistical power is limited owing to its small sample size and single-hospital setting, especially there are only nine patients in the unsuppressed PRA group and multivariate analysis could not be performed. In this study, we strictly selected patients with unsuppressed PRA because PRA is known to fluctuate [55]. This criterion might have led to the small sample size, but the use of this extreme and reproducible criterion could ensure the scientific validity of the study. As individuals with APA without PRA suppression are rare, multi-institutional studies are warranted to overcome this limitation. Second, the study employed a retrospective design, and the time ordering between variables at baseline was unclear. Third, some patients with unsuppressed PRA who were excluded from our study owing to the lack of surgical treatment and pathological examination might have had APA; identifying such patients requires the development of a more accurate screening method. Fourth, we cannot rule out the possibility of unmeasured confounders that might explain the observed association. For example, although there was no significant difference in the BMI within the two groups, some of the patients in the unsuppressed PRA group exhbited a higher BMI that could lead to sleep apnea syndrome (SAS). SAS could, in turn, lead to the elevation of PRA and cortisol levels via dysregulation of the renin-angiotensin-aldosterone system through altered sympathetic nervous system activation [56, 57]. Actually, some patients in the unsuppressed PRA group showed a high apnea-hypo index, as shown in S1 Table.

Here we described nine patients with PA with associated unsuppressed PRA in detail. In conclusion, our findings show that unsuppressed PRA should not be a clinical indicator to

rule out a PA diagnosis, particularly when a patient's cortisol levels are not suppressed in a DST. Furthermore, patients with PA with associated unsuppressed PRA may be at a higher risk for an incomplete cure of their hypertension after surgery. Multi-center investigations with larger sample sizes are needed to validate our findings and to identify additional characteristics of patients with PA with unsuppressed PRA.

## Supporting information

**S1 Fig. Histological HE and CYP11B2-staining images of aldosterone-producing adenomas with unsuppressed PRA.**
(TIF)

**S1 Table. Baseline clinical characteristics of each patient in the unsuppressed PRA group.**
(DOCX)

**S2 Table. Endocrine parameters of each patient in the unsuppressed PRA group.**
(DOCX)

**S3 Table. Results of segment-selective adrenocorticotropic hormone-loading adrenal venous sampling in the unsuppressed PRA group.**
(DOCX)

**S4 Table. Cohort dataset of the unsuppressed and suppressed PRA group.**
(XLSX)

## Acknowledgments

We thank a native English speaker for checking the language and grammar used in this manuscript (Editage, Tokyo, Japan).

## Author Contributions

**Conceptualization:** Haremaru Kubo, Yuya Tsurutani.

**Data curation:** Haremaru Kubo, Yuya Tsurutani, Kosuke Inoue, Kazuki Watanabe, Yuto Yamazaki, Takashi Sunouchi, Yoshitomo Hoshino, Rei Hirose, Sho Katsuragawa, Hiromitsu Tannai, Yukiko Shibahara, Yukio Kakuta, Seishi Matsui, Jun Saito, Masao Omura.

**Formal analysis:** Haremaru Kubo.

**Investigation:** Haremaru Kubo.

**Methodology:** Haremaru Kubo, Yuya Tsurutani, Kosuke Inoue, Hironobu Sasano.

**Project administration:** Haremaru Kubo.

**Supervision:** Yuya Tsurutani, Kosuke Inoue, Jun Saito, Masao Omura, Hironobu Sasano, Tetsuo Nishikawa.

**Validation:** Haremaru Kubo.

**Visualization:** Haremaru Kubo, Yuto Yamazaki.

**Writing – original draft:** Haremaru Kubo.

**Writing – review & editing:** Haremaru Kubo, Yuya Tsurutani, Kosuke Inoue, Tetsuo Nishikawa.

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
