## [Editor Report · Decision Letter 0]

25 Oct 2021

PONE-D-21-24186Characteristics of Aldosterone-producing Adenomas in Patients Without Plasma Renin Activity SuppressionPLOS ONE

Dear Dr. Tsurutani,

Thank you for submitting your manuscript to PLOS ONE. After careful consideration, we feel that it has merit but does not fully meet PLOS ONE’s publication criteria as it currently stands. Therefore, we invite you to submit a revised version of the manuscript that addresses the points raised during the review process.

We look forward to receiving your revised manuscript.

Kind regards,

Ali S. Alzahrani

Academic Editor

PLOS ONE

Journal Requirements:

2. During the internal evaluation of the manuscript it is in our understanding that   immunohistochemistry was performed on tissue samples of patients as a part of the study. As such please could you provide additional information regarding whether informed consent were taken from the participants in this case. Please ensure that you have specified the type of informed consent (ie verbal/written). If verbal, please specify how this was documented.

Editor Comments:

In this study, the authors compared 9 patients with primary aldosteronism (PA) and unsuppressed plasma renin activity (PRA) with 86 PA patients with suppressed PRA. All patients underwent selective adrenal vein sampling (AVS) and adrenal adenoma surgical removal. The histopathological examination showed adrenal adenomas with positive CYP11B2 staining. Differences between the two groups were shown with the unsuppressed PRA group having higher basal morning and suppressed cortisol levels, the higher aldosterone level in the contralateral adrenals on AVS, stronger CYP17 staining and higher rate of DM. The authors suggest that unsuppressed plasma renin activity should not be taken as evidence of absence of PA in the right context and that these cases might be related to higher autonomous cortisol secretion and that this group of patients with unsuppressed PRA may constitute a distinct phenotype of PA.

The concept is rational and the manuscript is detailed and well written. There are some commented that need to be addressed as follows:

1. Line 38, taken adrenalectomy, please rephrase

2. Line 104, you need to name and briefly describe the immunoassays or at least reference them

3. Line 146, the number of cases here seems to be wrong. Please correct

4. Line 228, please briefly describe what you mean by complete and partial success

5. Line 282, suppressed here probably meant to be unsuppressed, please chack

6. Line 288, might in the unresected side, please rephrase

7. Line 310, suppressed probably meant to be unsuppressed, please check
---

## [Author Response · Author response to Decision Letter 0]

10 Nov 2021

Editor comments: 

1. Line 38, taken adrenalectomy, please rephrase

Response: We rephrased as “underwent adrenalectomy” to match the grammar (Page 3, Line 33) and revised the manuscript accordingly.

2. Line 104, you need to name and briefly describe the immunoassays or at least reference them

Response: In order to describe the details of immunoassays of PACs, serum cortisol concentrations, and PRAs, we added the references of previous published relevant studies including standardized guidelines in Japan at Material and Methods Section as follows (Page 5, Lines 104-105) for clarification. We modified the sentence as “radioimmunoassay as previously described [11, 12] and standardly used [9, 10, 13].”

3. Line 146, the number of cases here seems to be wrong. Please correct

Response: We must apologize for this confusing sentence. In our hospital, 969 patients with PA visited our hospital while 2007 and 2020. Within these patients, because of the development and introduction of various up-dated systems, 9 and 86 patients fulfilled the inclusion criteria as unsuppressed and suppressed PRA in this study, respectively.

To clarify the method of the population selection, we modified our sentence as “Among patients with PA visited our hospital while 2007 and 2020, 9 and 86 patients fulfilled the inclusion criteria as unsuppressed and suppressed PRA in this study, respectively.” at Result Section (Page 7, Lines 146-147).

4. Line 228, please briefly describe what you mean by complete and partial success

Response: 

Response: 

We added the explanation phrase as “This standardized outcome criteria mean remission (complete success), improvement (partial success), and persistence (absent success) of PA after adrenalectomy from the viewpoint of both clinical (mainly about blood pressure and the existence of medications) and biochemical outcomes (mainly about excess of aldosterone and renin suppression), respectively.” at Results Section (Pages 11-12, Lines 251-255) and revised manuscript accordingly.

5. Line 282, suppressed here probably meant to be unsuppressed, please check

Response: We agreed with your comments. In this sentence we meant “The duration of hypertension was relatively longer in some cases in unsuppressed group (suppressed group, 10 [8, 19] vs. unsuppressed group, 8 [2, 16] years, P=0.15), but there is no significant difference between two groups.” To clarify our meaning, we fixed our sentence as “However, there were no significant differences in these values between two groups…” at Discussion Section (Page 14, Lines 307-308) and revised manuscript accordingly.

Additionally, we replaced up-dated reference at Discussion Section (Page 13, Lines 289-292) because the previous report was old with language restrictions in Japanese.

6. Line 288, might in the unresected side, please rephrase

Response: We have rephrased the term as suggested at Discussion Section (Page 14, Lines 313).

7. Line 310, suppressed probably meant to be unsuppressed, please check 

Response: Thank you for your careful review. We therefore replaced the word with “unsuppressed” (Page 14, Lines 338-339) and revised manuscript accordingly.

---

## [Decision Letter · Decision Letter 1]

24 Feb 2022

PONE-D-21-24186R1Characteristics of Aldosterone-producing Adenomas in Patients Without Plasma Renin Activity SuppressionPLOS ONE

Dear Dr. Tsurutani,

Thank you for submitting your manuscript to PLOS ONE. After careful consideration, we feel that it has merit but does not fully meet PLOS ONE’s publication criteria as it currently stands. Therefore, we invite you to submit a revised version of the manuscript that addresses the points raised during the review process.

ACADEMIC EDITOR:  Further review of the manuscript showed some important points that need to be addressed (please see reviewers' comments)

We look forward to receiving your revised manuscript.

Kind regards,

Ali S. Alzahrani

Academic Editor

PLOS ONE

Reviewers' comments:

Reviewer's Responses to Questions

**Comments to the Author**

1. If the authors have adequately addressed your comments raised in a previous round of review and you feel that this manuscript is now acceptable for publication, you may indicate that here to bypass the “Comments to the Author” section, enter your conflict of interest statement in the “Confidential to Editor” section, and submit your "Accept" recommendation.

Reviewer #1: (No Response)

Reviewer #2: (No Response)

2. Is the manuscript technically sound, and do the data support the conclusions?

Reviewer #1: Partly

Reviewer #2: Yes

3. Has the statistical analysis been performed appropriately and rigorously? 

Reviewer #1: Yes

Reviewer #2: Yes

4. Have the authors made all data underlying the findings in their manuscript fully available?

Reviewer #1: Yes

Reviewer #2: Yes

5. Is the manuscript presented in an intelligible fashion and written in standard English?

Reviewer #1: Yes

Reviewer #2: Yes

6. Review Comments to the Author

Reviewer #1: This is a very interesting retrospective study on the effect of unsuppressed renin levels on clinical outcome of patients with primary aldosteronism (PA) undergoing adrenalectomy (ADX). The manuscript itself is well written. The authors conclude that unsuppressed renin levels do not exclude even unilateral PA. However, complete clinical success after ADX is less likely in these cases.

I have the following comments:

1) In the methods section authors are stating that some patients ‘skipped some confirmatory tests’. What does this mean? Did every patient undergo a confirmatory test? What did authors do if two confirmatory tests showed discordant results? On which guidelines was the diagnosis based (JES? ESE?). Please clarify.

2) In line 149 authors describe four patients who did not fulfill screening criteria for ARR. Moreover, some of these patients had no pathological screening test either. Did those patients finally belong to the unsuppressed PRA group? Could this have had an impact on the analysis/outcome?

3) Again in the methods section authors state that ‘biochemical outcomes are excluded’. As we know from several studies, clinical outcome is important but has several confounding factors such as age, duration of hypertension, renal function and sex. This is why PASO criteria recommends assessing biochemical outcome to confirm ‘biochemical cure’ after 6-12 months using ARR or confirmatory testing. Please further clarify this issue.

4) A major concern is the small number of patients with unsuppressed renin levels enrolled in the study. Although the results are in part statistically significant, the sample size is very small. Moreover, patients with unsuppressed PRA showed significantly higher rates of diabetes mellitus. Could this have had an impact on the analysis? The authors may consider cumulating some patients and redoing the analysis.

5) Suppression of renin and outcome of PA have gained much attention during the past. Besides aldosterone, the amount of dietary salt intake plays a major role for the suppression of renin levels. This is why I am wondering whether authors also assessed 24-hour urinary sodium excretion to address this problem.

6) Please add the BP measurement methods to the methods section and refer to the manuscript of Lenders JWM et al. JCEM 2020. This section should be described more in details.

Reviewer #2: The authors investigated the clinical characteristics of PA patients with unsuppressed PRA compared to those with suppressed PRA. They found that both morning cortisol levels and cortisol levels after overnight dexamethasone suppression test in PA group with unsuppressed PRA were higher than with suppressed PRA. They also showed that success rate of clinical outcomes after 1 year in PA group with unsuppressed PRA was lower than with suppressed PRA because aldosterone levels at non-surgical side during AVS in PA group with unsuppressed PRA were higher than with suppressed PRA. They concluded that PA should not rule out by hypertensive patients with unsuppressed PRA, particularly when they have cortisol elevation. These findings are clinically valuable. However, there are several concerns about this manuscript.

Major comments

1. Why did the authors use “an Endocrine Society Clinical Practice Guideline” for the diagnosis of subclinical Cushing’s syndrome (2010) but not “New diagnostic criteria of adrenal subclinical Cushing’s syndrome: opinion from the Japan Endocrine Society” (2018) ?

2. The author should show the pathological data about the atrophy of the attached normal adrenal cortex after removal of the adrenal tumor in 5 PA patients with unsuppressed PRA who had cortisol levels more than 1.8 �g/dL after overnight dexamethasone suppression test.

3. The authors concluded that PA should not rule out by hypertensive patients with unsuppressed PRA, particularly when they have cortisol elevation. This conclusion is misleading. Each cortisol levels in 9 PA patients with unsuppressed PRA was within normal range. Cortisol elevation might mean autonomous cortisol secretion. However, there were not autonomous cortisol secretion in some PA patients with suppressed PRA.

For example, ACTH levels at 8:00 was 59.2 pg/mL and those at 23:00 was 8.7 pg/mL in Patient 8. These data suggest that ACTH was not suppressed by autonomous cortisol secretion and have diurnal change. How do they explain? They should correct a manuscript.

4. There is a possibility that overnight dexamethasone suppression test is false positive in patients with untreated sleep apnea syndrome. BMI in 4 PA patients with unsuppressed PRA was more than 25. Did the authors check sleep apnea syndrome in these patients?

5. The sympathetic nervous activation as well as the decrease in intravascular blood volume increase plasma renin activity. The sympathetic nervous activation by untreated sleep apnea syndrome might affect plasma renin activity. The authors should consider this possibility of the mechanism of unsuppressed PRA in addition to autonomous cortisol secretion.

6. The number of PA patients with unsuppressed PRA is few and clinical characteristics of them were also diverse (for example PAC [240 min after saline loading] 14.3-255.0 ng/dL). The authors should discuss the clinical characteristics of each PA patients with unsuppressed PRA.

7. PLOS authors have the option to publish the peer review history of their article (what does this mean?). If published, this will include your full peer review and any attached files.

Reviewer #1: No

Reviewer #2: No

---

## [Author Response · Author response to Decision Letter 1]

9 Apr 2022

Point-by-Point Responses to the Comments:

Dear Editors and Reviewers

Thank you very much for reviewing our manuscript and offering valuable advice.

We have read each of your comments carefully and provided our point-by-point responses below. The manuscript has been revised accordingly, as noted by the page and line numbers.

Additionally, we used Editage (www.editage.com) for English language editing such as grammar and spelling check of the manuscript.

Reviewer #1: 

1) In the methods section authors are stating that some patients ‘skipped some confirmatory tests’. What does this mean? Did every patient undergo a confirmatory test? What did authors do if two confirmatory tests showed discordant results? On which guidelines was the diagnosis based (JES? ESE?). Please clarify.

Reply:

We appreciate your insightful comments. What we intended to convey with this sentence was that we did not conduct saline-loading tests or furosemide loading tests for the patients who would be expected to have an elevated risk of complications from the tests, as previously reported (Eur J Endocrinol. 2019 Feb 1;180(2):R45–R58.)”.

We have described the results of the confirmatory tests in the Table S2. More specifically, one patient was exempted from the saline-loading tests and four patients were exempted from the furosemide loading tests in the unsuppressed PRA group because of severe hypokalemia, a prior history of stroke, or severe hypertension.

We usually diagnose PA based on the JES guidelines in clinical practice for those with one or more positive results in the confirmatory tests with suppressed PRA levels. Meanwhile, in this study, we screened for patients with unsuppressed PRA, which made it difficult for them to meet the diagnostic criteria for PA. Therefore, a diagnosis of PA was not one of the inclusion criteria in this study, as described in our response to the next question.

For the same reason, we did not include the JES guideline for a PA diagnosis as an inclusion criterion in the unsuppressed PRA group, as described in the Methods section (lines 111–125). In particular, we noted that four patients did not meet the criteria for a diagnosis of PA (lines 166–171 and 174–176). Moreover, the text describing the difficulty of diagnosing PA in the patients with unsuppressed PRA has been modified and expanded upon (lines 364–389).

2) In line 149 authors describe four patients who did not fulfill screening criteria for ARR. Moreover, some of these patients had no pathological screening test either. Did those patients finally belong to the unsuppressed PRA group? Could this have had an impact on the analysis/outcome?

Reply:

In the unsuppressed PRA group (nine patients), four patients exhibited an ARR < 20 (Table S2), one of whom did not have a positive result in the confirmatory tests.

The study’s inclusion criteria are described in the Methods section (lines 76–89). In this study, we regarded the presence of pathologically confirmed aldosterone-producing adenoma (APA) as the most important factor. Otherwise, we did not add the ARR values or the results of the confirmatory tests to the inclusion criteria because unsuppressed PRA itself decreases the ARR. However, all eight of the patients who completed the saline loading tests (which does not include PRA in its criterion) in the unsuppressed PRA group showed a positive result (lines 373–375), suggesting hyperaldosteronsim does exist. One patient (patient 3) was exempted from the saline loading test due to the risk of severe hypertension, as described in the response to the previous comment.

We emphasize that the take-home message of this study is the difficulty in diagnosing PA in those with unsuppressed PRA. Therefore, confirmatory tests that include PRA are not suitable inclusion criteria for the unsuppressed PRA group, and we used the presence of pathologically confirmed APA as an important inclusion criterion.

Even if we were to exclude patient 3, who did not have a confirmed positive result in the confirmatory tests, the main result would not differ. Incidentally, patient 3 did fulfill the study’s inclusion criteria, including the presence of pathologically confirmed APA.

There explanations have been added on lines 111–125, 166–171, 366–368, and 375-378.

3) Again in the methods section authors state that ‘biochemical outcomes are excluded’. As we know from several studies, clinical outcome is important but has several confounding factors such as age, duration of hypertension, renal function and sex. This is why PASO criteria recommends assessing biochemical outcome to confirm ‘biochemical cure’ after 6-12 months using ARR or confirmatory testing. Please further clarify this issue.

Reply:

In this study, we believed that ARR could not be used as an adequate parameter for assessing post-operative outcomes, as previously described. One important reason for this is that the biochemical outcome assessed in the PASO study requires ARR normalization; however, some patients in this study already exhibited a normal ARR even before surgery due to elevated PRA. Therefore, we would not have been able to assess the normalization of ARR in those patients, and we excluded this assessment of biochemical outcomes in this study.

However, biochemical outcomes can also include assessments of the correction of hypokalemia. In terms of hypokalemia, improvements were observed in the present study. Indeed, patients 1, 2, 6, 7, 8, and 9 (i.e., all of the patients exhibiting pre-existing hypokalemia), avoided the need for oral potassium supplementation after surgery. Therefore, a partial biochemical improvement might be suggested in these patients that is distinct from ARR normalization. We have added this explanation to lines 282–285 and 357–361.

4) A major concern is the small number of patients with unsuppressed renin levels enrolled in the study. Although the results are in part statistically significant, the sample size is very small. Moreover, patients with unsuppressed PRA showed significantly higher rates of diabetes mellitus. Could this have had an impact on the analysis? The authors may consider cumulating some patients and redoing the analysis.

Reply:

As described in the Materials and Methods section, we were very strict in our selection of patients in the unsuppressed PRA group, which included those with PRA levels of > 1.0 ng/mL/hr measured in any two or more blood samples due to known fluctuations in PRA (J Clin Endocrinol Metab 2003 Jun;88(6):2489–94.). We believe that this severe criterion ensured the scientific validity of the study, despite leading to the small sample size. Indeed, some patients exhibited inconsistent PRA levels across samples, such as the 0.8, 0.7, and 1.5 ng/dL/hr values measured in one patient (in this case, the patient was excluded due to the lack of reproducibility), as mentioned in the Materials and Methods section. To better explain this, we have added sentences on lines 391–394. Besides, only nine patients fulfilled these criteria of the 969 inpatients with suspected PA who were treated at our hospital between 2007 and 2020. This supports the fact that it is difficult to identify and include patients with unsuppressed PRA in studies of this type.

Regarding the influence of diabetes, diabetes itself could be a risk factor for the acceleration of processes that lead to atherosclerosis and chronic kidney disease. The existence of elevated PRA is known to be associated with atherosclerosis and cardiovascular disease in patients with diabetes (Diabetes Care. 2020 Apr;43(4):843–851.). Additionally, the patients with diabetes show the activation of the renin-angiotensin system and the elevation of PRA via chronic kidney disease (Horm Metab Res. 2013 May;45(5):338–43.). Additional text describing this association has been added on lines 327–332.

5) Suppression of renin and outcome of PA have gained much attention during the past. Besides aldosterone, the amount of dietary salt intake plays a major role for the suppression of renin levels. This is why I am wondering whether authors also assessed 24-hour urinary sodium excretion to address this problem.

Reply:

Thank you for your comment. To address this concern, we have added data on the 24-hr sodium urine levels in Table 1 and Table S1. There was no difference in sodium excretion between the two groups, probably due to the fact that patients with suspected PA were on a non-sodium restricted diet (routinely 8 g/day) in our hospital. This is described on lines 103–105 and 335–337.

6) Please add the BP measurement methods to the methods section and refer to the manuscript of Lenders JWM et al. JCEM 2020. This section should be described more in details.

Reply:

In our hospital, all blood pressure measurements were conducted by the attending physician to ensure the precise and adequate collection of clinical data (J Clin Endocrinol Metab. 2020 Jun 1;105(6):dgaa159.). In particular, the measurements were conducted according to the guidelines of The Japanese Society of Hypertension (Hypertens Res. 2019 Sep;42(9):1235–1481.). We have added text describing these assessments on lines 105–108.

Reviewer #2: 

1. Why did the authors use “an Endocrine Society Clinical Practice Guideline” for the diagnosis of subclinical Cushing’s syndrome (2010) but not “New diagnostic criteria of adrenal subclinical Cushing’s syndrome: opinion from the Japan Endocrine Society” (2018) ?

Reply:

We apologize for not updating the citation on the diagnosis of subclinical Cushing’s syndrome. We have revised the citation on line 179.

2. The author should show the pathological data about the atrophy of the attached normal adrenal cortex after removal of the adrenal tumor in 5 PA patients with unsuppressed PRA who had cortisol levels more than 1.8 �g/dL after overnight dexamethasone suppression test.

3. The authors concluded that PA should not rule out by hypertensive patients with unsuppressed PRA, particularly when they have cortisol elevation. This conclusion is misleading. Each cortisol levels in 9 PA patients with unsuppressed PRA was within normal range. Cortisol elevation might mean autonomous cortisol secretion. However, there were not autonomous cortisol secretion in some PA patients with suppressed PRA.

For example, ACTH levels at 8:00 was 59.2 pg/mL and those at 23:00 was 8.7 pg/mL in Patient 8. These data suggest that ACTH was not suppressed by autonomous cortisol secretion and have diurnal change. How do they explain? They should correct a manuscript.

Reply:

Thank you for providing the meaningful advice in Comments 2 and 3. 

Pathologically, there were no findings suggestive of the presence of apparent atrophy of the attached normal cortex in the unsuppressed PRA group. A previous study (Eur J Endocrinol. 2011 Apr;164(4):447–55.) reported that only 37.1% of aldosterone and cortisol co-secreting adrenal tumors demonstrated evidence of cortical atrophy in the adjacent non-neoplastic tissue. Therefore, patients with aldosterone and cortisol co-secreting adrenal tumors may display unique clinical and endocrinological features. Based on these reports, we have added additional sentences describing the tissue pathology on lines 245–246.

In addition, as Reviewer 2 noted, there were some patients in the unsuppressed PRA group who still maintained diurnal variation. Therefore, the wording of “when they have cortisol elevation” has been revised and alternative phrasing referring to “higher cortisol levels in the 1 mg DST” is now used. Also, the sentences about the patients who did not fulfill the criteria for SCS (Endocr J. 2018 Apr 26;65(4):383-393.) have been modified based on the guideline. The wording in the abstract has also been revised accordingly (line 46–47).

Regarding the diurnal variation, not all of the patients with a positive result in the 1 mg DST (cortisol level ≥1.8 �g/dL) exhibited ACTH suppression and a lack of diurnal variation. Some patients with these endocrinological features have been described even in older SCS guidelines (Endocr J. 2013;60(7):903–12.). A case report about a patient with SCS in whom diurnal variation was maintained has also been published (Endocr J. 2006 Oct;53(5):609–13.). Considering the findings of these reports, those with SCS or patients with a higher cortisol level in the 1 mg DST represent a clinically, pathologically, and endocrinologically heterogenous phenotypes. We have added text describing these findings on lines 296–297, 312, 314–315, 318–325, 326, and 411–412.

4. There is a possibility that overnight dexamethasone suppression test is false positive in patients with untreated sleep apnea syndrome. BMI in 4 PA patients with unsuppressed PRA was more than 25. Did the authors check sleep apnea syndrome in these patients?

5. The sympathetic nervous activation as well as the decrease in intravascular blood volume increase plasma renin activity. The sympathetic nervous activation by untreated sleep apnea syndrome might affect plasma renin activity. The authors should consider this possibility of the mechanism of unsuppressed PRA in addition to autonomous cortisol secretion.

Reply:

We are grateful for the suggestions and feedback provided in Comments 4 and 5. 

We have added data on apnea-hypopnea index (AHI) measurements in Table S1 and additional sentences describing the association between sleep apnea syndrome (SAS) (Sleep. 2009 Dec;32(12):1589–92.), PRA, and cortisol (Clin Endocrinol (Oxf). 2021 Dec;95(6):909–917.) on lines 401–407. In particular, some patients demonstrated a higher AHI, but not all of the patients had undergone screening for SAS, making it difficult to precisely evaluate the relationship (lines 402–408).

6. The number of PA patients with unsuppressed PRA is few and clinical characteristics of them were also diverse (for example PAC [240 min after saline loading] 14.3-255.0 ng/dL). The authors should discuss the clinical characteristics of each PA patients with unsuppressed PRA.

Reply:

The diverse clinical characteristics of the patients comprising the unsuppressed PRA group suggest a high degree of heterogeneity. Actually, only six of the patients exhibited higher cortisol levels after the 1 mg DST. Other known PRA-elevating factors are a lower eGFR indicative of chronic kidney disease (eGFR < 60 mL/min/1.73 m2), which was observed in three patients, and diabetes, which was present in five patients. Patient 2, in particular, had a prior history of stroke and a long duration of hypertension (21 years).

These multiple factors could have contributed to the elevation of PRA, and further investigations, including a higher number of patients, might elucidate the associations of each factor. Additional details describing the influence of such factors on PRA and PAC have been added to the manuscript (lines 380–389）.

Thank you very much for your thoughtful consideration. We hope we have addressed your concerns satisfactorily. We believe this topic will be of interest to your readers worldwide, and we look forward to your editorial decision. 

Please let us know if you have questions or additional suggestions. Thank you. 

Yuya Tsurutani

Endocrinology and Diabetes Center, Yokohama Rosai Hospital, Japan 3211 Kozukue-cho, Kouhoku-ku, Yokohama, Kanagawa, 222-0036, Japan. 

Tel: +81-45-474-8111 

Fax: +81-45-474-8323 

e-mail: yuya97tsuru1055@gmail.com

---

## [Editor Report · Decision Letter 2]

14 Apr 2022

Characteristics of Aldosterone-producing Adenomas in Patients Without Plasma Renin Activity Suppression

PONE-D-21-24186R2

Dear Dr. Tsurutani ,

We’re pleased to inform you that your manuscript has been judged scientifically suitable for publication and will be formally accepted for publication once it meets all outstanding technical requirements.

Kind regards,

Ali S. Alzahrani

Academic Editor

PLOS ONE

---

## [Editor Report · Acceptance letter]

20 Apr 2022

PONE-D-21-24186R2 

Characteristics of Aldosterone-producing Adenomas in Patients Without Plasma Renin Activity Suppression 

Dear Dr. Tsurutani:

I'm pleased to inform you that your manuscript has been deemed suitable for publication in PLOS ONE. Congratulations! Your manuscript is now with our production department. 

Kind regards, 

on behalf of

Dr. Ali S. Alzahrani 

Academic Editor

PLOS ONE